# Understanding the Robustness-Accuracy Tradeoff by Rethinking Robust Fairness

## Abstract

Although current adversarial training (AT) methods can effectively improve the robustness on adversarial examples, they usually lead to a decrease in accuracy, called the robustness-accuracy trade-off. In addition, researchers have recently discovered a robust fairness phenomenon in the AT model; that is, not all categories of the dataset have experienced a serious decline in accuracy with the introduction of AT methods. In this paper, we explore the relationship between the robustness-accuracy tradeoff and robust fairness for the first time. Empirically, we have found that AT will cause a substantial increase in the inter-class similarity, which could be the root cause of these two phenomena. We argue that the label smoothing (LS) is more than a trick in AT. The smoothness learned from LS can help reduce the excessive inter-class similarity caused by AT, and also reduce the intra-class variance, thereby significantly improving accuracy. Then, we explored the effect of another classic smoothing regularizer, namely, the maximum entropy (ME), and we have found ME can also help reduce both inter-class similarity and intra-class variance. Additionally, we revealed that TRADES actually implies the function of ME, which can explain why TRADES usually performs better than PGD-AT on robustness. Finally, we proposed the maximum entropy PGD-AT (ME-AT) and the maximum entropy TRADES (ME-TRADES), and experimental results show that our methods can significantly mitigate both tradeoff and robust fairness.

## 1 Introduction

### 1.1 Background

Deep neural networks (DNNs) have been proven to be vulnerable to the adversarial attacks, as demonstrated in (Szegedy; Goodfellow et al.; Kurakin et al.; Carlini & Wagner). By adding crafted imperceptible perturbations to the input, attackers can easily fool the model to give an incorrect prediction. To defend against adversarial attacks, tens of methods have been proposed, but most of them later proved to be ineffective (Athalye et al., 2018). Among these many defense techniques, adversarial training (AT) (Madry et al., 2017) has been proven to be the most effective strategy against adversarial attacks.

Although current AT algorithms can effectively improve model robustness, there are two confusing phenomena in AT models. First, there can be an inevitable robustness-accuracy tradeoff (Tsipras et al., 2018) in AT models in which increasing robustness is always accompanied by an accuracy drop. Second, recently Xu et al. (2021) found that AT tends to introduce severe disparities in accuracy and robustness between different classes. For example, as shown in Figure 1b, in a PGD-AT model (Madry et al., 2017), both the accuracy and robustness of the 3rd class *cat* are much lower than those of the 1st class *car*, while the two classes have similar accuracies in the standard training model (see Figure 1a). This phenomenon is defined as the robust fairness according to the authors.

Additionally, as Xu et al. (2021) mentioned, the robust fairness problem is closely related to the robustness-accuracy tradeoff, because the average accuracy drop in the robustness-accuracy tradeoff could mainly come from the classes that are hard to classify in AT. To verify this, we have measured the accuracy drop for each class, and calculated their percentage in the total accuracy drop, as shown in Figure 1c. We can see that it only takes two classes (the *cat* and *bird*) to contribute almost half of

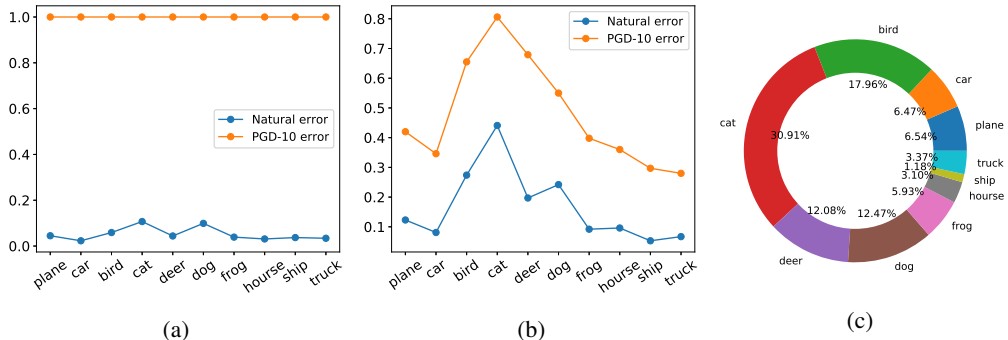

Figure 1: (a): Classwise natural/PGD-10 errors of standard training model. (b): Classwise natural/PGD-10 errors of PGD-AT model. (c): Contribution of each class in the total accuracy drop.

the accuracy drop, while the two classes have the lowest accuracy and robustness than other classes in AT. That is, these hard classified classes have a significantly greater impact on the decline in accuracy, and to better understand the robustness-accuracy tradeoff, it should be determined why these classes are so difficult to classify in AT.

To explain the phenomenon, Xu et al. (2021) argued that some classes are difficult to classify in AT because they are intrinsically "harder" to classify than other classes, and AT tends to hurt both accuracy and robustness of these "hard" classes. To verify this point of view, these authors studied the effect of AT on a binary classification task under a mixture Gaussian distribution, and the "hard" class is the one with larger variance in their case. They showed that AT will push the decision boundary closer to the larger variance class and further worsen both the accuracy and robustness of the class.

However, although they showed that the class with a larger variance is more difficult to classify, there still remains a question; that is, is variance enough to describe the "hard" degree of a class? Imagine two Gaussian distributions both have a high variance, but also an extremely large difference of mean values, they should still be well classified. On the contrary, when the two Gaussian distributions both have a low variance, but the mean values of them are extremely similar, which makes the two distributions severely overlap, we cannot satisfactorily classify them instead. That is, the inter-class similarity is also an important factor affecting the model's accuracy. With this point in mind, we have measured both inter-class similarity and intra-class variance in standard training, PGD-AT and TRADES models (Zhang et al., 2019) for each class in the CIFAR10 test set, as shown in Figure 2.

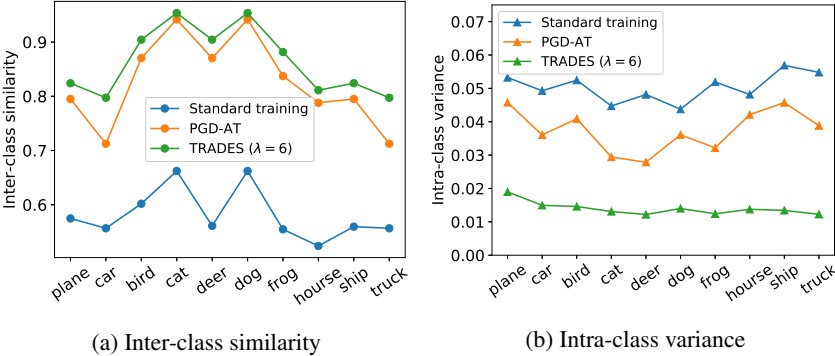

(a) Inter-class similarity      (b) Intra-class variance

Figure 2: Measurement of the inter-class similarity and intra-class variance in standard training, PGD-AT and TRADES models.

The measurement is performed in the penultimate layer feature space. For each class, we use the variance of features as the class's intra-class variance. To measure the inter-class similarity of each

class, we first calculate the feature mean value vectors for all classes, and then the cosine similarity between the mean value vectors of the measured class and other classes. The largest cosine similarity is used as the inter-class similarity of the measured class in this paper. It is somewhat surprising to see that both PGD-AT and TRADES models have a lower variance than the standard training model in Figure 2b, while they have a worse accuracy instead. However, as shown in Figure 2a, both PGD-AT and TRADES can lead to a higher inter-class similarity than standard training. In particular, we notice that the "hardest" class *cat* does not have the largest variance no matter in PGD-AT, TRADES or the standard training model, but has the largest inter-class similarity. These observations have challenged Xu et al. (2021)'s theory that the "hard" classes are the large variance classes and indicate that inter-class similarity does matter in AT, thus motivated us to study both robust fairness phenomenon and robustness-accuracy tradeoff toward the increased inter-class similarity.

### 1.2 OUR CONTRIBUTIONS

**Understand the robustness-accuracy tradeoff & robust fairness.** To the best of our knowledge, we are the first to study the relationship between the robustness-accuracy tradeoff and the robust fairness, and we find that the two phenomena could both come from the increased inter-class similarity caused by AT. More specifically, through our single AT and binary classification AT experiments in section 2, we find that:

- AT will cause a general increase in inter-class similarity for each class, which even causes a feature overlap, and finally leads to the accuracy drop in the tradeoff.
- The "hard" classes in AT are actually similar classes in standard training, and the increased inter-class similarity in AT makes them more similar and harder to be classified, which causes the robust fairness problem.

**Re-investigate the effect of smoothing regularizer in AT.** Label smoothing (LS) (Szegedy et al., 2016) has been used as a trick to benchmark the robustness in AT by Pang et al. (2020), however, we noticed that LS can not only help improve the robustness, but usually improve the accuracy too, which means a reduction in the robustness-accuracy tradeoff. In this paper, we find LS can help alleviate the tradeoff because it helps reduce the large inter-class similarity in AT, and also provides a lower intra-class variance. Then, we investigate the effect of the maximum entropy (ME) Pereyra et al. (2017), which is also a classic smoothing regularizer, and we find ME can help reduce both inter-class similarity and intra-class variance too. In addition, we find that the state-of-the-art AT method TRADES can be seen as a special maximum entropy learning, which could explain why TRADES model have a lower intra-class variance than PGD-AT model in Figure 2, and usually performs better than PGD-AT in terms of robustness. We proposed the maximum entropy PGD-AT (ME-AT) and the maximum entropy TRADES (ME-TRADES), and experimental results show that our methods can significantly mitigate both tradeoff and robust fairness.

## 2 RETHINKING ROBUST FAIRNESS TOWARD INTER-CLASS SIMILARITY

In Figure 2a, we have shown that AT models have higher inter-class similarity than the standard training model. In this section, we design two experiments to see how the high inter-class similarity in AT is related to both the robustness-accuracy tradeoff and robust fairness phenomena.

### 2.1 SINGLE ADVERSARIAL TRAINING

**AT causes a feature overlap.** We design the single adversarial training (single AT) to see how AT affect one single class. In single AT, we only conduct adversarial training on one class while training other classes normally. For better visualization, we adjust the penultimate layer of a ResNet-18 model to output 2-D features. In Figure 3, we show the single AT results of the two most representative classes: the "hardest" class *cat* and the "easiest" class *car*. The results of other classes and detailed settings are both provided in Appendix A.1. In Figure 3b, when single adversarial train the 3rd class *cat*, the features of the *cat* severely overlap with those of the 5th class *dog*, and the overlapping features make the class *cat* almost impossible to classify (only has 7.03 natural accuracy and 0 PGD-10 robustness). This observation intuitively shows how the inter-class similarity increases in

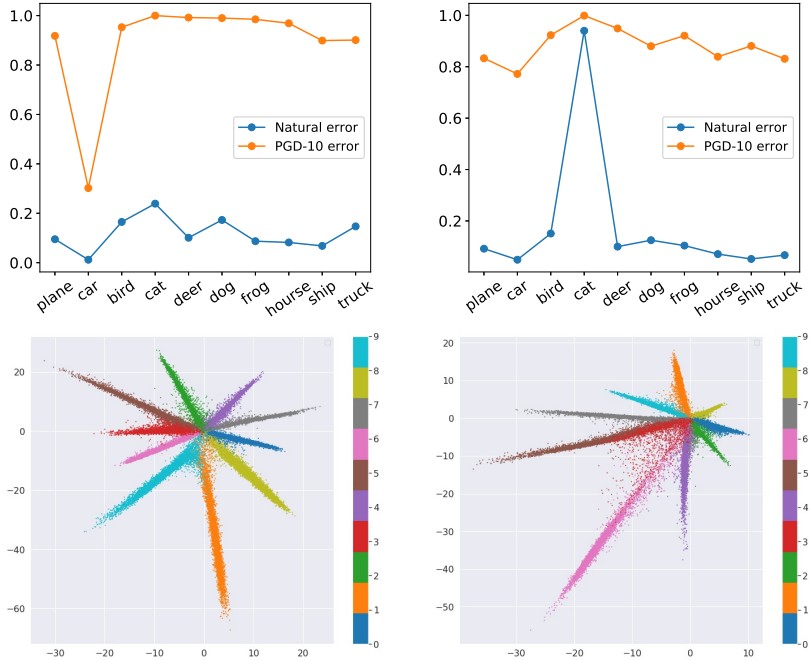

(a) Single adversarial train the 1st class *car*  (b) Single adversarial train the 3rd class *cat*

Figure 3: Results of the single AT experiment. Top line: the classwise errors; Bottom line: the 2-D feature representations.

AT, and proves that the accuracy drop part in the robustness-accuracy tradeoff could come from the increased inter-class similarity (the overlapping features).

**The increase in inter-class similarity is general in AT.** However, when single AT is carried out for the 1st class *car*, the features of the class *car* can still be split well from other classes, and both the accuracy and PGD-10 robustness of the class *car* achieve a high level (98.4 and 72.2 respectively, see Figure 3a). Does this mean that the "easy" classes can avoid an increase in inter-class similarity in AT?

To check this, we measure the inter-class similarity in the single AT models, and for comparison, we also measured the inter-class similarity of a standard training 2-D features ResNet-18 model. As shown in Figure 4, each class in the blue line represents the inter-class similarity of the class in the corresponding single AT model (e.g., the point of the class *car* in the blue line represents the inter-class similarity of the class *car* in the single *car* AT model), and the yellow line is the inter-class similarity of the standard training model. We can see that even the "easiest" class *car* in the single AT model can have a higher inter-class similarity than that in the standard training model. This observation turns out that the increase in inter-class similarity is general in AT for all classes.

## 2.2 BINARY CLASSIFICATION ADVERSARIAL TRAINING

**"Hard" classes or similar classes?** Since the increase in inter-class similarity is general for all classes, we assume that some classes are difficult to classify in AT possibly because they are already similar in standard training, and the increased inter-class similarity caused by AT makes them more similar and become the "hard" classes. To verify this assumption, we conduct the binary classification AT experiments. We set the class *cat* to binary classify with other classes in CIFAR10 dataset, and we use both PGD-AT (Madry et al., 2017) and TRADES (Zhang et al., 2019) to train our binary classification ResNet-18 models (512-D features here).

We plot the natural error and PGD-10 error of the PGD-AT and TRADES trained binary classification models in Figure 5a and Figure 5b respectively. Classes in the horizontal axis represent the classes that binary classified with the class *cat*, and is sorted from small to large by their similarity

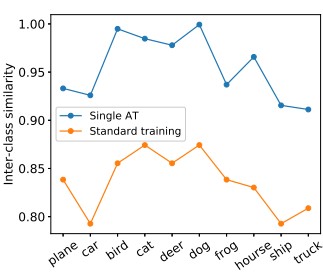 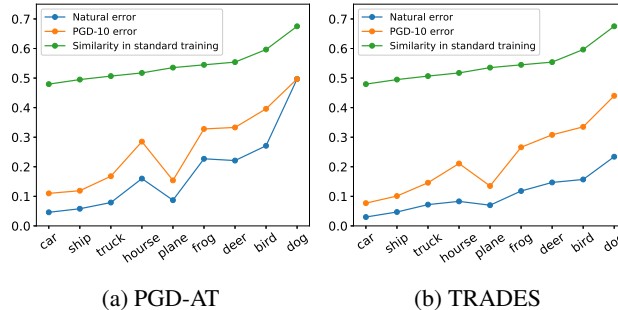

                                                     (a) PGD-AT                (b) TRADES

Figure 4: Inter-class similarity of standard training model and single AT models.

Figure 5: (a) and (b) show the natural/PGD-10 error of binary classification PGD-AT and TRADES models respectively. Similarity with the class *cat* in standard training is also plotted at the green line to see the correlation.

with the *cat* in standard training. We find that both natural error and PGD-10 error in the binary classification PGD-AT and TRADES models are highly positive correlated with the similarity in standard training. For example, the class *car* is the least similar class of the *cat* in standard training, when binary classified *cat* with the *car*, model can get both low natural error and PGD-10 error (4.6 and 11.0); However, when binary classified cat with the most similar class dog, the ResNet-18 model even failed to converge in PGD-AT (49.7 for both natural and PGD-10 error), and even model can converge in TRADES, it is also in both highest natural error and PGD-10 error (23.4 and 44.0). This observation indicates that the "hard" classes in AT could actually be the similar classes in standard training.

### 2.3 UNDERSTANDING THE TRADEOFF & ROBUST FAIRNESS

To briefly summarize, through our single AT and binary classification AT experiments, we find the following:

- AT will even cause a feature overlap to the "hard" classes, which leads to a severe accuracy drop.

- The increase in inter-class similarity is general in AT for all classes.

- "Hard" classes in AT may actually be similar classes in standard training for generally increased inter-class similarity.

These findings indicate that the increased inter-class similarity could be the root cause of both robustness-accuracy tradeoff and robust fairness problems, and indicate a new way to mitigate the tradeoff: To obtain better robustness and accuracy, the excessive inter-class similarity in AT should be reduced (while not increasing the intra-class variance). And in the next section, we show this way is promising by the effect of smoothing regularizer.

**Our explanation.** Finally, we provide an intuitive explanation for why AT leads to higher inter-class similarity. The core object of AT is to force adversarial examples under the same distribution as the well-classified clean examples. To achieve this object, TRADES directly minimizes the kl-divergence between adversarial and clean examples and minimizes the cross-entropy loss of clean examples to achieve high accuracy. In PGD-AT, this object is implicit. PGD-AT directly minimizes the cross-entropy loss of adversarial examples, and because adversarial examples can be seen as a robust lower bound of clean examples, it actually forces both adversarial examples and clean examples to fit the same one-hot label. However, forcing adversarial examples to fit the distribution of clean examples may also lead clean examples to be closer to the distribution of adversarial examples. While adversarial examples are naturally closer to other classes, the adoption of adversarial examples could pull classes closer, then resulting in higher inter-class similarity.

# 3 THE ROLE OF THE SMOOTHING REGULARIZER

Based on our findings in section 2, in this section, we argue that LS is not only a trick in AT. We find that smoothness learned from LS can significantly reduce both inter-class similarity and intra-class variance in AT, which is the remedy to current AT methods. Then, we investigate the effect of the ME, which is also a classic smoothing regularizer, and we find TRADES can be seen as a special ME learning. Finally, we proposed the ME-AT and ME-TRADES to mitigate both robustness-accuracy tradeoff and robust fairness.

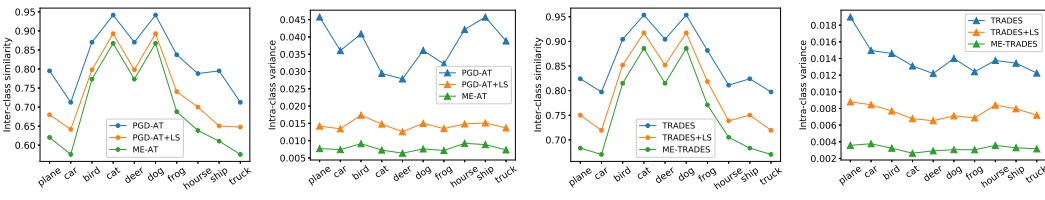

| (a) Similarity in PGD-AT | (b) Variance in PGD-AT | (c) Similarity in TRADES | (d) Variance in TRADES |

Figure 6: LS and ME can help reduce both the inter-class similarity and intra-class variance in PGD-AT and TRADES.

## 3.1 LABEL SMOOTHING IS NOT ONLY A TRICK IN AT

LS has been recently used as a trick to benchmark the robustness by Pang et al. (2020). However, we noticed that LS can usually help improve accuracy too, which means a reduction in the robustness-accuracy tradeoff. In this paper, we find LS can help mitigate the tradeoff for reasons. By visualizing the penultimate layer feature representations of standard models with/without LS in a 2-D figure, Müller et al. (2019) showed that LS encourages the features of training examples from the same class to group in tighter clusters. That is, *LS could help reduce the inter-class similarity and inrta-class variance in standard training.* To see if LS has the same effect in AT, we measured the inter-class similarity and intra-class variance in the PGD-AT and TRADES models with/without LS. As shown in Figure 6, we find LS can indeed reduce both similarity and variance.

Therefore, we argue that LS is more than a trick in AT. Compared to standard training, LS is more significant in AT for reducing the excessive inter-class similarity of robust models. As a result, in Table 1, we can see that LS can more significantly improve accuracy in both the PGD-AT and TRADES models than in the standard training model. We also measured the robustness under AutoAttack (AA) (Croce & Hein, 2020), which is currently the most effective adversarial attack, and we can see that LS also increases AA accuracy in PGD-AT by 1.12.

Table 1: LS ($\alpha = 0.1$) helps improve more accuracy in PGD-AT and TRADES than in standard training (ST).

|  | ST+LS | ST | Diff | PGD-AT+LS | PGD-AT | Diff | TRADES+LS | TRADES | Diff |
|---|---|---|---|---|---|---|---|---|---|
| Natural Accuracy | 94.87 | 94.82 | +0.05 | 84.00 | 83.35 | **+0.65** | 82.74 | 82.43 | **+0.31** |
| AA | 0 | 0 | 0 | 48.76 | 47.64 | **+1.12** | 49.75 | 49.95 | -0.2 |

However, we find that when adding LS into TRADES, the AA robustness becomes 0.2 lower than the original TRADES model, as shown in Table 1. This could happen because LS will cause a loss of information in the logits, and hence weaken the discriminative power of the trained models (Müller et al., 2019), which could hurt the performance of semi-supervised learning (knowledge distillation in Müller et al. (2019)'s case and TRADES here). This pitfall of LS motivates us to investigate the effect of another classic smoothing regularizer, namely, the ME.

### 3.2 UNDERSTANDING THE EFFECT OF MAXIMUM ENTROPY

#### 3.2.1 PROPOSED METHOD

**ME learning.** Let us first take a brief look at the ME learning. Let $p_\theta(\boldsymbol{x})$ be the probability distribution of input $\boldsymbol{x}$ produced by a DNN model. The entropy of this conditional distribution is given by:

$$H(p_\theta(\boldsymbol{x})) = -\sum_i p_\theta(\boldsymbol{x})_i \log p_\theta(\boldsymbol{x})_i$$

By adding the negative entropy term into the cross-entropy loss during training, the object of ME learning is defined as:

$$\mathcal{L}_{ME}(\theta) = CE(\boldsymbol{x}, \boldsymbol{y}) - \beta H\left(p_\theta(\boldsymbol{x})\right), (\beta > 0)$$

where $\boldsymbol{y}$ represents the one-hot labels, and $CE$ is the cross-entropy function:

$$CE(\boldsymbol{x}, \boldsymbol{y}) = -\sum_i \boldsymbol{y}_i \log p_\theta(\boldsymbol{x})_i$$

**TRADES is a special ME learning.** Then, we show that the sota AT method TRADES is also a ME learning. Let us recall that the objective function of TRADES is:

$$\mathcal{L}_{TRADES}(\theta) = CE(\boldsymbol{x}, \boldsymbol{y}) + \lambda \cdot KL(p_\theta(\boldsymbol{x})\|p_\theta(\boldsymbol{x}'))$$

where $x'$ is the adversarial example, and $KL$ is the kl-divergence, which is given by:

$$KL(p_\theta(\boldsymbol{x})\|p_\theta(\boldsymbol{x}')) = \sum_i p_\theta(\boldsymbol{x})_i \log p_\theta(\boldsymbol{x})_i - \sum_i p_\theta(\boldsymbol{x})_i \log p_\theta(\boldsymbol{x}')_i$$

$$= -H(p_\theta(\boldsymbol{x})) - \sum_i p_\theta(\boldsymbol{x})_i \log p_\theta(\boldsymbol{x}')_i$$

And the TRADES objective function can be rewritten as:

$$\mathcal{L}_{TRADES}(\theta) = \underbrace{CE(\boldsymbol{x}, \boldsymbol{y}) - \lambda H(p_\theta(\boldsymbol{x}))}_{\text{maximum entropy learning}} \underbrace{- \lambda \sum_i p_\theta(\boldsymbol{x})_i \log p_\theta(\boldsymbol{x}')_i}_{\text{adversarial cross-entropy}}$$

We can see that the left part corresponds to ME learning, and the right part is a cross-entropy loss between the distribution of clean and adversarial examples. This finding reveals that TRADES is a special ME learning, as the most direct result, we can see that TRADES model have larger entropy than PGD-AT model on both clean and adversarial examples in Figure 7. Then, we notice that even the entropy value of the TRADES model is still far from the limit entropy value in a 10-classification condition (approximately 2.3), which means that there should be enough space for the PGD-AT and TRADES models to receive a stronger ME regularization. Based on this fact, we proposed the maximum entropy PGD-AT (ME-AT) and maximum entropy TRADES (ME-TRADES).

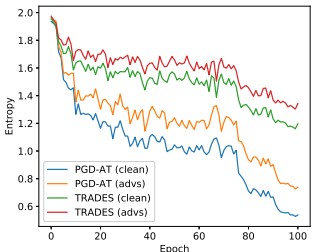

Figure 7: Entropy in PGD-AT and TRADES of each epoch on CIFAR10 test set.

**ME-AT & ME-TRADES.** Here, we formulize the objective function of ME-AT and ME-TRADES. For ME-AT, we maximize the entropy of adversarial example distribution and we have the object:

$$\mathcal{L}_{ME-AT}(\theta) = CE(\boldsymbol{x}', \boldsymbol{y}) - \beta H\left(p_\theta(\boldsymbol{x}')\right), (\beta > 0)$$

For ME-TRADES, we augment the ME hyperparameter of clean example distribution in TRADES, and the object is:

$$\mathcal{L}_{ME-TRADES}(\theta) = CE(\boldsymbol{x}, \boldsymbol{y}) - (\lambda + \beta) H(p_\theta(\boldsymbol{x})) - \lambda \sum_i p_\theta(\boldsymbol{x})_i \log p_\theta(\boldsymbol{x}')_i, (\beta > 0)$$

and for code simplicity , we realize ME-TRADES in a more concise way as:

$$\mathcal{L}_{ME-TRADES}(\theta) = CE(\boldsymbol{x}, \boldsymbol{y}) + \lambda \cdot KL(p_\theta(\boldsymbol{x})\|p_\theta(\boldsymbol{x}')) - \beta H(p_\theta(\boldsymbol{x}))$$

which only needs to add a negative entropy term into the original TRADES code.

### 3.2.2 EXPERIMENT

**Training setting.** Our experiments are based on CIFAR10, which is the most popular dataset in AT. We perform the standard CIFAR10 data augmentation: a random 4 pixel crop followed by a random horizontal flip. We train ResNet-18 for 100 epochs using SGD with 0.9 momentum, and the batch size is 64. The initial learning rate is 0.1 and reduced to 0.01 and 0.001 at epochs 75 and 90, respectively. The weight decay is $2 \times 10^{-4}$. We use the 10-step PGD adversary in training, and we set the perturbation size $\epsilon = 0.03125$ under the $\ell_\infty$ norm and the step size is fixed to 0.008.

**Test setting.** To evaluate robustness, we use PGD-20, C&W-20 (Carlini & Wagner, 2017) and AA (Croce & Hein, 2020) to generate adversarial examples at $\epsilon = 0.03125$ under the $\ell_\infty$ norm too. We report the test accuracy/robustness of the best checkpoint that achieves the highest robustness under PGD-20 on the test set.

**Reduce inter-class similarity & intra-class variance.** We first measure the inter-class similarity and intra-class variance of our ME-AT and ME-TRADES models. As shown in Figure 6, we can see that ME can more effectively help reduce both similarity and variance compared with LS. While TRADES is also a ME learning method, this effect of ME can explain why TRADES model have lower variance than PGD-AT model in Figure 2b. Note that the inter-class similarity of the TRADES model is little higher than that in the PGD-AT model in Figure 2a because the adversarial regularization hyperparameter $\lambda = 6$ corresponds to a strong adversarial regularization that makes the similarity higher. To clear see the effect of $\lambda$ in terms of inter-class similarity and intra-class variance, we measured both inter-class similarity and intra-class variance at $\lambda = 1, 6, 12$ in TRADES, and when $\lambda = 1$, TRADES is in both lower similarity and variance than PGD-AT (see Figure 9 in the Appendix).

**Enable larger hyperparameter** $\lambda$. Then, we evaluate the performance of TRADES and our ME-TRADES, as shown in Table 2. We find that ME-TRADES can adopt a higher adversarial regularization hyperparameter $\lambda$ than the original TRADES, and we can see that when $\lambda = 6$, the robustness of TRADES reaches the maximum (49.95 AA accuracy). Then, the robustness decreases as the $\lambda$ increases. However, we find that ME-TRADES reaches the maximum robustness when $\lambda = 27$, which is much larger than the maximum robustness parameter $\lambda = 6$ in the original TRADES, and the maximum robust accuracy is also higher (50.61 AA accuracy). This could be caused by the effect of ME that effectively reduces the inter-class similarity and intra-class variance, which could improve the *effective capacity* of the DNN model and allow the model to receive a stronger adversarial regularization, then help achieve higher robustness. Therefore, this effect of ME may explain why TRADES usually performs better than PGD-AT on robustness.

Table 2: Evaluation results of TRADES and ME-TRADES.

| Method | Natural | PGD-20 | C&W-20 | AA |
|---|---|---|---|---|
| TRADES ($\lambda = 6$) | 82.43 | 53.14 | 51.00 | 49.95 |
| TRADES ($\lambda = 9$) | 79.61 | 53.49 | 50.76 | 49.83 |
| TRADES ($\lambda = 12$) | 78.26 | 53.56 | 50.65 | 49.90 |
| TRADES ($\lambda = 15$) | 77.01 | 53.15 | 50.13 | 49.33 |
| ME-TRADES ($\beta = 1, \lambda = 15$) | **83.11** | 54.33 | 51.18 | 49.94 |
| ME-TRADES ($\beta = 1, \lambda = 18$) | 82.08 | 54.36 | 51.11 | 50.04 |
| ME-TRADES ($\beta = 1, \lambda = 24$) | 80.67 | 54.79 | 51.21 | 50.24 |
| ME-TRADES ($\beta = 1, \lambda = 27$) | 80.49 | **54.76** | **51.50** | **50.61** |

**Mitigate the robustness-accuracy tradeoff.** While the large model capacity has been shown crucial for reducing the robustness-accuracy tradeoff (Madry et al., 2017), we find that this possible improvement in effective capacity from ME can also help mitigate the tradeoff. We show the ME-AT results in Table 3. When the ME hyperparameter $\beta = 0.5$, ME-AT performs better in terms of both robustness and accuracy than PGD-AT; therefore, ME can effectively mitigate the tradeoff.

**Mitigate the robust fairness.** In order to check whether ME-AT and ME-TRADES can mitigate the robust fairness problem, we follow Xu et al. (2021)'s setting that uses the worst class performance to measure the fairness, where a higher worst class accuracy/robustness means better fairness. How-

Table 3: Evaluation results of PGD-AT, PGD-AT+LS and ME-AT.

| Method | Natural | PGD-20 | CW-20 | AA |
|---|---|---|---|---|
| PGD-AT | 83.35 | 50.86 | 50.16 | 47.64 |
| PGD-AT+LS($\alpha = 0.1$) | 84.00 | 52.09 | **50.8** | **48.76** |
| ME-AT($\beta = 0.5$) | **84.34** | 52.31 | 50.43 | 48.16 |
| ME-AT($\beta = 1$) | 82.44 | **53.09** | 49.94 | 48.16 |

ever, we are not going to compare with the fair robust learning algorithm (FRL) proposed by Xu et al. (2021). FRL is proposed to solve the fairness problem, which can significantly increase the worst class performance; however, it also causes a worse average performance than the previous sota AT method TRADES. In contrast, in this paper, our ME-AT and ME-TRADES are first proposed to mitigate both robustness-accuracy tradeoff and robust fairness, i.e., increase the worst class performance and average performance at the same time, which is a more difficult task than only improving the worst class performance. Therefore, it is not appropriate to compare FRL to our method. As shown in Table 4, both ME and LS can help increase the worst class accuracy and robustness when added to PGD-AT. Because TRADES already contains the ME, we can see that TRADES performs better than PGD-AT with respect to fairness. We show the results of ME-TRADES when $\beta = 1$ and $\lambda = 15, 27$, which correspond the largest average accuracy and robustness parameters, respectively. ME-TRADES can further improve the worst class accuracy ($\lambda = 15$) and robustness ($\lambda = 27$) compared to TRADES.

Table 4: Worst class natural accuracy & PGD-20,CW-20 and AA robustness.

| Method | Natural | PGD-20 | CW-20 | AA |
|---|---|---|---|---|
| PGD-AT | 55.9 | 18.7 | 17.2 | 13.7 |
| PGD-AT+LS | 63.9 | **25.3** | **23.6** | **20.4** |
| ME-AT($\beta = 0.5$) | **65.0** | 25.0 | 22.9 | 19.7 |
| TRADES ($\lambda = 6$) | 65.0 | 26.8 | 23.7 | 21.8 |
| TRADES+LS ($\lambda = 6$) | 64.4 | 25.7 | 22.4 | 20.8 |
| ME-TRADES ($\beta = 1, \lambda = 15$) | **65.6** | 25.7 | 22.5 | 21.1 |
| ME-TRADES ($\beta = 1, \lambda = 27$) | 63.0 | **29.0** | **24.4** | **23.0** |

**Benchmark the AA leaderboard.** To benchmark the robustness on the AA leaderboard, we combine our method with the robust self-training (RST) (Carmon et al., 2019), which adds 500K preprocessed data into training and obtains a value of 60.37 in the AA robustness at $\epsilon = 8/255$ on CIFAR10. Training details are provided in Appendix A.3.1.

## 4 CONCLUSION

In this paper, we corroborate that AT will cause an increase in inter-class similarity, which could be the root of both the robustness-accuracy tradeoff and robust fairness phenomena. We confirm that ME can help reduce the excessive inter-class similarity in robust models, and also provides a lower intra-class variance, which is the remedy of previous AT methods. Our work could provide new insight into understanding and mitigating both the robustness-accuracy tradeoff and robust fairness.

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

# A  APPENDIX

## A.1  TRAINING DETAILS AND RESULTS OF SINGLE AT EXPERIMENTS

**Training details of single AT.** Our single AT experiments performed on the ResNet-18 model. A linear layer (512×2) was added before the fc layer in the ResNet-18 model to output 2-D features, and the weight size of the fc layer was adjusted as (2×10) to output the logits. We set bias = None for the fc layer, which means we can estimate the similarity of two classes only from their included angle in the 2-D feature representations figure. For the single adversarial trained class, we perform a PGD-AT (Madry et al., 2017) at perturbation size $\epsilon = 0.03125$ under the $\ell_\infty$ norm, and for other classes we perform the standard training. The single AT results of the 1st class *car* and the 3rd class *cat* has been shown in Figure 3, the results of the remaining classes are provided in Figure 8 below.

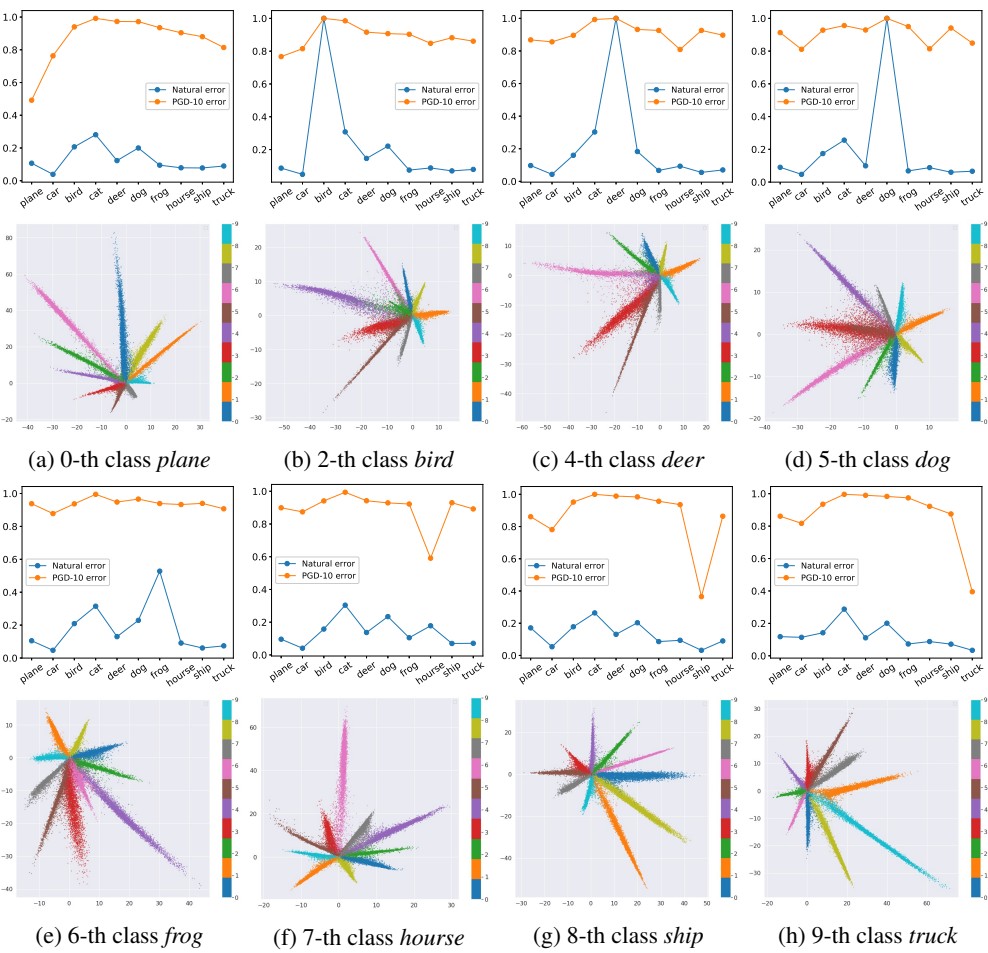

Figure 8: Results of single AT of the 0,2,4,5,6,7,8,9 classes in CIFAR10 test set. We can see the severe feature overlap also happens in 8b,8c,8d.

## A.2  LARGER $\lambda$ CAUSES HIGHER INTER-CLASS SIMILARITY

In Figure 9, we can see that with the adversarial regularization hyperparameter $\lambda$ increases, the inter-class similarity also increases. And because of the effect of ME that reducing the intra-class variance, we also find the intra-class variance will decrease when the hyperparameter $\lambda$ increases.

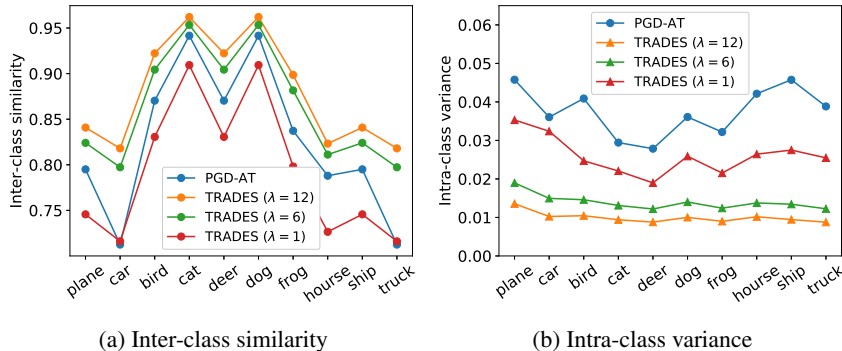

(a) Inter-class similarity                    (b) Intra-class variance

Figure 9: Measurement of the inter-class similarity and intra-class variance of TRADES ($\lambda = 1, 6, 12$) and PGD-AT.

### A.3 BENCHMARK THE AA LEADERBOARD

#### A.3.1 ME-RST

Carmon et al. (2019) used the additional pre-processed 500K data to train robust models, which is the RST procedure. To benchmark robustness in the AA (Croce & Hein, 2020) leaderboard, we combine the maximum entropy regularizer with the RST method (the ME-RST). To boost performance, we replaced the batch normalization (BN) layer to the batch-instance normalization (BIN) layer (Nam & Kim, 2018)in the WideResNet-28-10 model, reasons see A.3.2. We only change the $\lambda$ to 27 (which is the highest robustness parameter in our ResNet-18 experiments) in the original RST settings[1], and the ME hyperparameter $\beta$ is set to be 1. Our results are tested at the perturbation size $\epsilon = 8/255$ under the $\ell_\infty$ norm. We compare our method with the top-6 methods on the robustbench website[2] which also perform on the WideResNet-28-10 model and use $\epsilon = 8/255$ for testing robustness. Evaluation results are shown in Table 5.

Table 5: AutoAttack performance on CIFAR10, using WideResNet-28-10 model under $\epsilon = 8/255$.

| Method/Paper | Natural Accuracy | AA |
|---|---|---|
| Gowal et al. (2020) | 89.48 | 62.8 |
| Rade & Moosavi-Dezfooli (2021) | 88.16 | 60.97 |
| Rebuffi et al. (2021) | 87.33 | 60.75 |
| Wu et al. (2020) | 88.25 | 60.04 |
| Sridhar et al. (2021) | 89.46 | 59.66 |
| RST (Carmon et al., 2019) | 89.69 | 59.53 |
| ME-RST | 88.51 | 59.96 |
| ME-RST+BIN | **89.11** | **60.37** |

#### A.3.2 ROBUST MODEL MAY NEED TO LEARN MORE SHAPE FEATURES

Inspired by our findings in this paper, we pose an open question: How does DNNs recognize two classes as similar in standard training? Because if we know that, then we can design a more robust model architecture with low inter-class similarity. However, this question is still a hard problem in deep learning, because it actually ask us to answer another question first: What kind of features does DNN models learned?

We noticed that a previous work has studied this question. Geirhos et al. (2018) showed that DNNs could be biased towards the texture feature, however loses the shape feature. This may explain why

---

[1]RST's github `https://github.com/yaircarmon/semisup-adv`
[2]The AA leaderboard `https://robustbench.github.io/`

the class *dog* and *cat* are so hard to classify in our binary experiment: *dog* and *cat* have the similar texture feature (the *hair*). Therefore, robust models may need to pay more attention to the shape feature. To learn more shape feature, we attempted to replace BN layer as the BIN layer, which is proposed by Nam & Kim (2018) to balance the shape and texture features, and experimental results indicate that BIN can effectively help improve both accuracy and robustness in Table 5.

