# OpenReview forum: "Understanding the robustness-accuracy tradeoff by rethinking robust fairness "
_ICLR.cc/2022/Conference — ICLR 2022 Submitted_

### Official Review · Reviewer_94sP · 2021-10-21

**Correctness:** 3
**Technical Novelty And Significance:** 2
**Empirical Novelty And Significance:** 3
**Recommendation:** 6
**Confidence:** 4

**Main Review:**

Strength:
1. The paper provides extensive experimental analysis (e.g., single adversarial training) for the potential reason of trade-off between robustness and accuracy and robust fairness. The paper concludes that high inter-class similarity might be the main cause.
2. The paper investigates the insight behind why TRADES performs better than PGD-AT.
3. The paper proposed new ME-based methods to further improve the adversarial robustness over baseline.

Weakness:
1. The improvement by ME is marginal compared with non-ME methods. Given that AA might not be the proxy of strongest adversarial attack, the true improvement might be smaller or even negative.
2. Regarding writing, in Section 3.1 the method of LS is not introduced. For those who are not familiar with LS, it is hard to understand the section.
3. The paper misses certain references which study the trade-off between robustness and accuracy, e.g., [1].
[1] A Closer Look at Accuracy vs. Robustness, NeurIPS 2020

**Summary Of The Paper:**

This paper investigates inter-class similarity and intra-class variance, and corroborates that AT will cause an increase in inter-class similarity, which could be the root of both the robustness-accuracy tradeoff and robust fairness phenomena. The authors first considers Label Smoothing (LS) as the regularizer, and concludes that LS will cause a loss of information in the logits, and hence weaken the discriminative power of the trained models. The authors then confirms that ME can help reduce the excessive inter-class similarity in robust models, and also provides a lower intra-class variance, which is the remedy of previous AT methods. Experiments partially support the conclusions of the paper.

**Summary Of The Review:**

Overall, I think the paper is on borderline in its current version. The main weakness of the paper is its technical novelty. But on the other hand, the paper provides extensive experimental results and a (potentially) reasonable explanation for the big thing in adversarial robustness: trade-off between robustness and accuracy and robust fairness. Extra theoretical analysis in support of the empirical discovery will significantly improve the paper and weight my rate towards recommending full acceptance.

---

### Official Review · Reviewer_EY7N · 2021-10-22

**Correctness:** 3
**Technical Novelty And Significance:** 1
**Empirical Novelty And Significance:** 2
**Recommendation:** 3
**Confidence:** 4

**Main Review:**

The paper starts with an empirical observation that large inter-class similarity highly correlates with the drop in accuracy as well as the accuracy parity between different classes. This observation is quite natural, since if the features of two classes are class to each other, then it makes it harder to classify for any predictors based on these features, leading to dropped accuracy of these classes. Motivated by this empirical observation, the authors proposed to use label smoothing (LS) to decrease the inter-class similarity, in order to alleviate the robustness-accuracy tradeoff problem.

Besides the label smoothing, the authors also argued that using the maximum entropy regularizer could help reduce the inter-class similarity, and then proposed two variants of existing works, i.e., adversarial training and TRADES, with the additional maximum entropy regularizer.

Overall the paper is quite clear and easy to follow. However, the current manuscript does not have a related work section and indeed fails to discuss a line of works in the fairness literature. In particular, the problem studied in this work is closely related to the accuracy parity problem in the fairness literature, yet there is no discussion in this line of works in the related works. Some of them include [1-3] and the references therein.

[1].    Demonstrating accuracy equity and predictive parity performance of the compas risk scales in broward county.

[2].    Understanding and Mitigating Accuracy Disparity in Regression

[3].    Fair regression: Quantitative definitions and reduction based algorithms.

The organization of the paper, however, could be improved. For example, Section 3 directly jumps to the discussion of label smoothing without a proper introduction on what is LS. The same also applies to the maximum entropy regularization as well.

Technically, although the empirical observation is fine, I am interested in understanding whether this phenomenon could generalize to other datasets, or is this specific to the one studied in this paper. To answer this question, it would be good to have some theoretical justification to link the relationship between robust accuracy and inter-class similarity. This is quite important, as it means the empirical observation so far is not just a coincidence but holds in general.

I would suggest the authors soften the use of the word "proven" in the description of the related works in the first paragraph of the Introduction. Technically, none of the existing methods has been "proven" to be the best but verified empirically to be effective. To me, claiming these empirical results to be proven is quite misleading.

Other questions:
-   In Fig. 2, I would image the features from the penultimate layer to be a high-dimensional vector, so the corresponding measure should be a covariance matrix rather than a scalar variance. What's the specific measure used in Fig. 2(a)?

Minor:
-   In the caption of Fig. 1, better to explain what's PGD-10 error, since this is not a standard term.


**Summary Of The Paper:**

Motivated by the empirical observations, the authors argue that there is a high correlation between the drop of robust accuracy, robust fairness and inter-class similarity. The authors then proposed to augment these two existing methods with the corresponding label smoothing and the maximum-entropy regularizer.


**Summary Of The Review:**

Given the lack of technical contributions and the discussion to a closely related line of works in the fairness literature, I would vote for rejection.

---

### Official Review · Reviewer_NCTP · 2021-10-26

**Correctness:** 3
**Technical Novelty And Significance:** 3
**Empirical Novelty And Significance:** 3
**Recommendation:** 3
**Confidence:** 4

**Details Of Ethics Concerns:**

No concerns.

**Main Review:**

<Strong points>

* Improving the fairness of robust training is a timely problem that is being actively studied.
* The empirical results on how adversarial training increases inter-class similarities, which reduces robust fairness, is convincing.
* The experiments show that ME-AT and ME-TRADES indeed improves the robust-accuracy tradeoff and robust fairness.

<Weak points>

* Overall the paper uses too many pages on motivating the problem while missing critical content as explained in the following weak points. The explanation on why AT algorithms increase inter-class similarities goes on for 5 pages, but the content is often redundant and can be significantly reduced while being just as convincing. The remaining 4 pages seem too short for the rest of the material.
* In the Introduction, the PGT-AT and TRADES techniques appear without much explanation. How do these techniques work and why are they the important ones to consider for fair robust training? Moreover, there is no related work section, and the authors should compare their techniques with the following recent fair robust training works:
   * Zhang and Davidson, "Towards Fair Deep Anomaly Detection", FAccT 2021.
   * Khani and Liang, "Removing Spurious Features can Hurt Accuracy and Affect Groups Disproportionately", FAccT 2021.
* Section 3.1 is critical where it explains why LS prevents inter-class similarities from going up. However, the explanation is not convincing where it only shows a few experiments on one dataset that is not even well described (is it the dataset mentioned in the Introduction?). As LS is an important technique, it should be explained in the paper instead of just adding a citation. Most importantly, there needs to be some convincing analysis showing why LS reduces similarity and variance for any dataset in general.
* The flow from Section 3.1 to 3.2 is not clear. If a few LS techniques work well empirically, why does that lead you to investigate ME? Do all LS techniques work well? Is there no LS technique that works better than ME? What if ME works well for different reasons? Among the ME methods, why only consider the TRADES method? If TRADES is already a special ME technique, how is ME-TRADES an improvement?
* The experiments are not extensive enough where only the CIFAR-10 dataset is used. Instead, there must be at least two or three datasets to make sure the results are general. The same comment goes for the experiments in the Introduction. In addition, the authors do not compare their techniques with FRL by Xu et al. because it is said to solve an easier problem and is thus "not appropriate". I am not sure if I agree. FRL is also one of the SOTA methods and has the similar goal of improving robustness against adversarial data while adding constraints to reduce accuracy disparities between groups. IMHO the authors should definitely make an extensive comparison with FRL to emphasize the effectiveness of ME-TRADES.
* In Section 3.2.2, please clearly define PGD-20 and the exact measure used to produce the Table 2 values.
* The role of intra-class variance is not clear. The inter-class similarity seems to be the main cause of accuracy-robustness tradeoff and robust fairness in the paper. In addition, existing AT methods already decrease the intra-class variance compared to standard training. Hence it is unclear how important intra-class variance is.


**Summary Of The Paper:**

This paper identifies how adversarial training (AT) algorithms for robustness may negatively affect the notion of robust fairness and proposes two methods ME-AT and ME-TRADES, which combine existing AT methods with a maximum entropy (ME) term, to improve the accuracy-robustness tradeoff and robustness fairness. Although AT algorithms improve model robustness, they also increase inter-class similarities, which make certain classes more difficult to classify, leading to unfair accuracies. The paper then shows that label smoothing (LS) mitigates this effect and in particular investigates the ME technique. The authors show that a method called TRADES outperforms another method called PGD-AT because it is a special version of ME. Experiments show that combining ME with these AT methods outperforms PGD-AT.

**Summary Of The Review:**

While the paper solves a timely and important problem, it can also be improved by shortening its introduction to the problem, better explaining why LS works and why we should focus on ME and TRADES, citing and comparing with more related work, and making its evaluations more extensive.

---

### Official Review · Reviewer_VwP1 · 2021-11-02

**Correctness:** 3
**Technical Novelty And Significance:** 3
**Empirical Novelty And Significance:** 3
**Recommendation:** 6
**Confidence:** 3

**Main Review:**

Strengths:
- The problem is well motivated, and the authors presented a very thorough analysis on the per-class accuracy and inter-class similarity on why AT could lead to a trade-off over accuracy and robustness. The reasoning train from section 1 through section 2 is interesting and well supported by experiments and analysis, and the method in section 3 is well motivated.

- The empirical results are strong on CIFAR-10 and demonstrate a better accuracy-robustness trade-off compared to two defense baselines (AT and TRADES).

Weakness:
- Currently all the results/analysis are on CIFAR-10 only. More experiment on other datasets would be useful to show the hypothesis (AT hurts inter-class similarity) holds across multiple scenarios.

- The authors should add more discussion on related works. A few works have shown that the improved adversarial robustness from label smoothing might be the effect of gradient-masking and thus LS could be volatile to other attacks. [1] Please see the discussion in: https://openreview.net/forum?id=BJlr0j0ctX; and [2] Fu et al. Label Smoothing and Adversarial Robustness. https://arxiv.org/pdf/2009.08233.pdf

Given the current paper claims more on the robustness-accuracy trade-off side, complete adversarial robustness might not be a goal, but it would be great if the authors can add more discussion on this front.

Minor:
- Figure 2 b, please pair the colors with class categories;
- Section 3.2.1, please add the citation back for ME learning (as in Section 1.2)
- Table 5 is informative and puts the results in perspective, consider moving it to main text.

**Summary Of The Paper:**

This paper investigates why adversarial training can sometimes present a worse trade-off between robustness and accuracy, and found one root cause could be that AT causes a substantial increase in the inter-class similarity. The paper then proposes combining both AT and TRADES with two smoothing techniques, label-smoothing, and maximum entropy, and found the resulting methods yield a better trade-off over natural accuracy and robustness.

**Summary Of The Review:**

This paper presents an interesting observation that adversarial training might decrease inter-class similarity, which might in turn hurt the robustness-accuracy trade-off. The proposed method that combines label smoothing, or maximum entropy, with both AT and TRADES, seems to improve the accuracy-tradeoff by quite a margin.

On the other hand, the authors should present more results on other datasets to show the results hold across scenarios. Also more discussion on related works would be useful to clarify the benefits of label smoothing in terms of adversarial robustness (it's known that LS improves accuracy).

---

### Decision · Program_Chairs · 2022-01-20

**Decision:**

Reject

**Comment:**

The paper argues that adversarial training increases inter-class similarities, therby increasingly the misclassification of some classes and lowering accuracy parity across classes. It proposes to combine existing adversarial training methodologies, PGD-AT and TRADES, with a maximum entropy term to improve the classification fairness while remaining robust.

While they agree that the problem is timely and important, the reviewers identify the following issues that place the current iteration of the paper below the bar of acceptance: the comparison to other works on fair robust training and accuracy parity is incomplete; experimental evaluation is conducted only on CIFAR10, making the generalizability of the paper's claims about performance unclear; and the proposed methodology has low technical novelty.